# Exploring the Binding of Natural Compounds to Cancer-Related G-Quadruplex Structures: From 9,10-Dihydrophenanthrenes to Their Dimeric and Glucoside Derivatives

**DOI:** 10.3390/ijms24097765

**Published:** 2023-04-24

**Authors:** Chiara Platella, Andrea Criscuolo, Claudia Riccardi, Rosa Gaglione, Angela Arciello, Domenica Musumeci, Marina DellaGreca, Daniela Montesarchio

**Affiliations:** 1Department of Chemical Sciences, University of Naples Federico II, Via Cintia 21, 80126 Naples, Italy; chiara.platella@unina.it (C.P.); andrea.criscuolo2@unina.it (A.C.); claudia.riccardi@unina.it (C.R.); rosa.gaglione@unina.it (R.G.); angela.arciello@unina.it (A.A.); domenica.musumeci@unina.it (D.M.); marina.dellagreca@unina.it (M.D.); 2Institute of Biostructures and Bioimages, CNR, 80134 Naples, Italy; 3CINMPIS—Consorzio Interuniversitario Nazionale di Ricerca in Metodologie e Processi Innovativi di Sintesi, Via E. Orabona 4, 70125 Bari, Italy

**Keywords:** G-quadruplex, natural compounds, cancer, dihydrophenanthrenoids, glucosides

## Abstract

In-depth studies on the interaction of natural compounds with cancer-related G-quadruplex structures have been undertaken only recently, despite their high potential as anticancer agents, especially due to their well-known and various bioactivities. In this frame, aiming at expanding the repertoire of natural compounds able to selectively recognize G-quadruplexes, and particularly focusing on phenanthrenoids, a mini-library including dimeric (**1**–**3**) and glucoside (**4**–**5**) analogues of 9,10-dihydrophenanthrenes, a related tetrahydropyrene glucoside (**6**) along with 9,10-dihydrophenanthrene **7** were investigated here by several biophysical techniques and molecular docking. Compounds **3** and **6** emerged as the most selective G-quadruplex ligands within the investigated series. These compounds proved to mainly target the grooves/flanking residues of the hybrid telomeric and parallel oncogenic G-quadruplex models exploiting hydrophobic, hydrogen bond and π-π interactions, without perturbing the main folds of the G-quadruplex structures. Notably, a binding preference was found for both ligands towards the hybrid telomeric G-quadruplex. Moreover, compounds **3** and **6** proved to be active on different human cancer cells in the low micromolar range. Overall, these compounds emerged as useful ligands able to target G-quadruplex structures, which are of interest as promising starting scaffolds for the design of analogues endowed with high and selective anticancer activity.

## 1. Introduction

DNA secondary structures, such as G-quadruplexes, are promising targets for anticancer strategies in the context of targeted therapies [1,2]. G-quadruplexes are essentially located at the end of human chromosomes, i.e., at telomeres, and in the promoter regions of genes coding for oncoproteins [3,4]. Several small-molecule ligands able to interact with these structures in vitro proved to be strong anticancer agents also in vivo, thus fully confirming the potential of therapeutic approaches based on G-quadruplex ligands [5,6,7].

A large variety of synthetic compounds have been widely explored as putative G-quadruplex binders [8,9,10]. In contrast, natural compounds have been far less studied, notwithstanding that they represent appealing candidates due to their different bioactivities and high structural diversity. On this basis, we recently focused on natural compounds aiming at extending the knowledge on the interaction of this class of molecules with G-quadruplexes [11,12,13,14,15,16]. A particular interest was devoted to 9,10-dihydrophenanthrene and dihydrodibenzoxepin derivatives isolated from Juncaceae, which were investigated for the first time in their ability to interact with G-quadruplex structures [14]. Among them, the dihydrodibenzoxepin derivative emerged as the strongest and most selective G-quadruplex ligand, showing anticancer activity on human cancer cells with an IC_50_ value of ~60 µM. In turn, dihydrophenanthrene derivatives showed lower affinity in their binding to the investigated telomeric and oncogenic G-quadruplex models [14].

Based on these results, this work was focused on the evaluation of dimeric and glucoside analogues of dihydrophenanthrenes aiming at identifying, within this class of compounds, chemical entities with improved G-quadruplex binding properties. Additionally, a related tetrahydropyrene glucoside was studied in parallel to evaluate the importance of the aglycone and sugar portions in these potential ligands in the interaction with G-quadruplex structures.

In detail, the library of compounds here investigated, reported in Table 1, included the dimeric phenanthrenoids **1**, **2** and **3**, the dihydrophenanthrene glucosides **4** and **5**, the tetrahydropyrene glucoside **6**, along with the 9,10-dihydrophenanthrene **7**, studied as a control.

All compounds were previously isolated from *Juncus effusus* and *Juncus acutus,* wetland Juncaceae of the Mediterranean area [17,18,19,20]. Particularly, the 9,10-dihydrophenanthrene **7** showed cytotoxic effects by the brine shrimp lethality assay (LC_50_ = 17.3 µM) [17]. The dimeric phenanthrenoids **1**, **2** and **3** showed antialgal activities (IC_50_ = 21.6, 16.7 and 12.3 µM, respectively), which were higher than those detected for the related 9,10-dihydrophenanthrenes [18,19]. The dihydrophenanthrene glucosides **4** and **5** were tested for their cytotoxic effects with the brine shrimp lethality assay and were found to be inactive at concentrations up to 120 µM. Thus, comparing these data with those of their aglycone **7**, it emerged that glycosylation drastically reduced the brine shrimp lethality of this class of compounds [17]. Finally, no data on the cytotoxicity of the tetrahydropyrene glucoside **6** [20] are present in the literature.

Herein, these natural compounds were studied for the first time as putative anticancer agents targeting human telomeric and oncogenic G-quadruplex-forming sequences. In parallel, their interaction with duplex-forming sequences used as controls was evaluated to assess the ability of the tested compounds to selectively target G-quadruplex structures compared to the duplex DNA. The DNA/ligand interactions were investigated by the affinity chromatography-based G4-CPG (G-quadruplex on Controlled Pore Glass) assay [21,22,23], circular dichroism (CD), fluorescence spectroscopy and molecular modelling. Finally, the strongest and most selective binders of the G-quadruplex models were tested for their antiproliferative activity on a panel of cancer cell lines.

## 2. Results and Discussion

### 2.1. G4-CPG Assay

The affinity chromatography-based G4-CPG assay was exploited for a quick and reliable screening of the examined natural compounds (Table 1), in order to evaluate their ability to target telomeric and oncogenic G-quadruplexes and discriminate them over duplex DNA [21,22,23]. This assay consists of flowing through a CPG support, functionalized with the target oligonucleotide sequence, each investigated ligand diluted in a washing solution, and evaluating the amount of the DNA-bound ligand by simple UV-vis analysis carried out after treating the support with a proper releasing solution, causing full G-quadruplex and duplex denaturation.

The natural compounds were first dissolved in DMSO to prepare stock solutions and then tested for their solubility in the washing/releasing aq. solution used for the assay [21,22,23]. All compounds proved to be fully soluble and stable in the assay solutions.

The sequence tel_26_ forming a hybrid G-quadruplex [24] and the sequence c-myc folding into a parallel G-quadruplex [25] in intracellular salt conditions were used as G-quadruplex models of the human telomeric and *c-myc* oncogene promoter DNA, respectively. On the other hand, the sequence ds_27_ forming a unimolecular hairpin duplex was used as the control duplex model [26].

Overall, in our assays **1** and **4** proved to be completely unable to interact with G-quadruplex- and duplex-functionalized supports, and **5** and **7** showed a null-to-very low affinity for all the functionalized supports (Table 2). On the other hand, **2** exhibited a medium-to-high affinity for G-quadruplex-functionalized supports, however with no significant difference compared to the duplex-functionalized support (Table 2). Conversely, **3** and **6** showed some, although modest, affinity towards the G-quadruplex-functionalized supports but were associated with a good G-quadruplex vs. duplex selectivity, since no ability to interact with the duplex-functionalized support was observed (Table 2). Therefore, **3** and **6** were selected as the most promising G-quadruplex selective ligands within the here examined series. Further studies on their interaction in solution with telomeric and oncogenic G-quadruplex models, as well as a control duplex, were thus carried out by CD and fluorescence spectroscopy.

### 2.2. Circular Dichroism Studies

The DNA sequences tel_26_ and Pu22 were chosen as models of telomeric and oncogene promoter G-quadruplexes respectively, while the sequence ds_12_, forming a bimolecular duplex well-mimicking the B-DNA conformation found in vivo [24,26], was used as control.

In the used experimental conditions, tel_26_ formed a hybrid 2-type G-quadruplex [27], as inferred by its CD spectrum showing a maximum at 290 nm, a shoulder at 265 nm and a minimum at 240 nm (Figure 1A, black line), while Pu22 folded into a parallel G-quadruplex [13,28,29,30], showing a CD spectrum with a maximum at 265 nm and a minimum at 240 nm (Figure 1B, black line). On the other hand, the CD profile of ds_12_ confirmed its structuring into a B-DNA conformation, being featured by a maximum at 280 nm and a minimum at 251 nm (Figure 1C, black line) [31]. As far as the thermal stability of the DNA models is concerned, melting temperature (T_m_) values of 43, 27 and 65 °C were found for the free tel_26_, Pu22 and ds_12_, respectively (Figure 1D–F, black lines).

After analysis of the free DNA systems, the selected oligonucleotide models were titrated with increasing amounts of **3** and **6** (up to 10 molar equivalents) and CD spectra were recorded after each addition (Figure 1A–C and Figure 2A–C, respectively).

In the case of the titration of tel_26_ G-quadruplex with **3**, slight variations of the intensity of the 290 nm positive band as well as of the 240 nm negative band were observed (Figure 1A), while no significant variation of the CD profile was found for tel_26_ G-quadruplex titrated with **6** (Figure 2A). As concerns the titrations of Pu22 G-quadruplex, a significant increase in the CD intensity of the 265 nm positive band as well as the 240 nm negative band was observed for both **3** and **6** (Figure 1B and Figure 2B). On the other hand, no relevant change was detected in the 280 nm and 251 nm bands of ds_12_ duplex for both ligands (Figure 1C and Figure 2C).

CD melting experiments were also performed in the presence of the two ligands for each DNA sequence. No relevant stabilizing effects were observed for any of the selected DNA models in the presence of **3** and for tel_26_ G-quadruplex and ds_12_ duplex in the presence of **6** (Figure 1D–F and Figure 2D,F, respectively), while stabilization of +4 °C was observed when Pu22 G-quadruplex was treated with **6** (Figure 2E).

Overall, CD titration experiments demonstrated that the main folds of the tel_26_ G-quadruplex and the control duplex were essentially preserved upon interaction with **3** and **6**, while significant effects were found on the Pu22 G-quadruplex fold in the presence of both ligands. Moreover, CD melting experiments proved that **6** exerted higher stabilizing properties than **3**, also associated with a higher G-quadruplex vs. duplex selectivity.

### 2.3. Fluorescence Spectroscopy Studies

In addition to CD-based studies of the DNA/ligand interactions, fluorescence spectroscopy analyses were carried out. Unfortunately, notwithstanding its full chemical stability over time as demonstrated by UV-vis analysis, a reduction in the fluorescence intensity of **6** with time was observed (in 30 min the fluorescence intensity of its maximum was ca. 10-fold lower compared to that of the freshly prepared solution, data not shown), thus precluding any possibility of studying the binding behaviour of this compound by fluorescence spectroscopy. In contrast, **3** showed a stable fluorescence over time and its spectrum was featured by a maximum at 375 nm and a shoulder at 450 nm (Figure 3, left panels, black lines).

Fluorescence titration experiments were thus performed on **3** by adding increasing amounts of tel_26_ G-quadruplex, Pu22 G-quadruplex and ds_12_ duplex to solutions of the ligand kept at a fixed concentration. A dose-dependent fluorescence quenching was detected in all titration experiments (Figure 3A–C), proving that **3** was able to interact with all the investigated DNA sequences.

Then, the elaboration of fluorescence data allowed obtaining the binding constants for all three DNA/ligand systems. In detail, the fraction of bound ligand was calculated from the fluorescence intensity maximum values and plotted as a function of the DNA concentration. These data were then fitted with an independent and equivalent-sites model (Figure 3D–F).

The highest binding constant was found for the interaction between the ligand and tel_26_ G-quadruplex [K_b_ = (2.3 ± 0.9) × 10^6^ M^−1^], while similar constant values were obtained for the Pu22 G-quadruplex [K_b_ = (4.4 ± 0.7) × 10^5^ M^−1^] and the ds_12_ duplex [K_b_ = (6.0 ± 1.7) × 10^5^ M^−1^], revealing a preference of the ligand for the hybrid G-quadruplex model. Indeed, the observed deviation of the fitting curves, obtained by the independent and equivalent-sites model, from the fluorescence titration experimental points can be explained considering that multiple binding events, each featured by a different binding constant, can occur in the interaction of **3** with the G-quadruplex models and the control duplex. Thus, the obtained binding constants should be considered as an average of the constants related to single binding events involving **3** and each of the studied DNA sequences.

### 2.4. Molecular Docking Studies

Molecular docking studies were carried out to get a deeper insight into the binding mode and interactions of **3** and **6** with tel_26_ G-quadruplex, Pu22 G-quadruplex and ds_12_ duplex.

As far as the interaction of **3** with tel_26_ G-quadruplex is concerned, the ligand showed as the preferred binding site the central region of the groove between the first and second G-rich tract of tel_26_ G-quadruplex, forming a hydrogen bond between the oxygen of OH-6 of **3** and the hydrogen of the exocyclic NH_2_ of G12 (Figure 4A). A binding energy of −8.3 kcal/mol was calculated for this binding pose. On the other hand, **6** targeted the free-loop groove of the tel_26_ G-quadruplex in the proximity of the 3′-end (Figure 4B). In this case, two hydrogen bonds were found, i.e., one between the oxygen of the OH-3′ of the glucose moiety of **6** and the hydrogen of the exocyclic NH_2_ of G23, and the other between the hydrogen of the glucose OH-4′ of **6** and the phosphate oxygen of T25. A binding energy of −8.2 kcal/mol was calculated for this binding pose.

Concerning the binding of **3** with the Pu22 G-quadruplex, the ligand was well accommodated in the free-loop groove of the Pu22 G-quadruplex in the proximity of the 3′-end (Figure 4C). A π-π interaction between the aromatic ring A of **3** and the guanine G6 was found. The binding energy for this binding pose was −7.1 kcal/mol. On the other hand, **6** targeted the 3′-end flanking residues of the Pu22 G-quadruplex (Figure 4D), with the formation of a hydrogen bond between the hydrogen of the OH-2 of pyrene moiety of **6** and the N1 of A21. A binding energy of −7.8 kcal/mol was calculated for this binding pose.

Finally, in the case of the interaction of **3** with the ds_12_ duplex, the most populated cluster showed a pose involving the binding of the ligand to the central region of the major groove of the duplex (Figure 4E), to which a binding energy of −8.0 kcal/mol was associated. A hydrogen bond between the oxygen of the OH-2′ of **3** and the hydrogen of the exocyclic NH_2_ of A5 was observed. On the other hand, **6** targeted one of the extremities of the minor groove of the duplex (Figure 4F). A binding energy of −7.9 kcal/mol was calculated for this binding pose.

Overall, the molecular docking analyses proved that both **3** and **6** mainly interacted with the grooves/flanking residues of the G-quadruplex and duplex models by hydrophobic, hydrogen bond and π-π interactions. In agreement with the fluorescence results, the highest binding energies were found for **3** interacting with the hybrid tel_26_ G-quadruplex, while similar binding energies were obtained for the parallel Pu22 G-quadruplex and the ds_12_ duplex. Interestingly, the two ligands showed similar binding energies between them in the recognition of both tel_26_ G-quadruplex and ds_12_ duplex, whereas in the case of the Pu22 G-quadruplex, a significantly higher binding energy was found for **6** compared to **3**, in agreement with CD thermal denaturation results.

### 2.5. Evaluation of the Cytotoxic Effects of **3** and **6** on Human Cancer Cell Lines

The antiproliferative activity on human cancer cells of **3** and **6** was determined by in vitro MTT assays on HeLa adenocarcinoma, MCF7 breast cancer and A431 epidermoid carcinoma cells. As shown in Figure 5 and Appendix A, both compounds were found to exert dose-dependent toxic effects on HeLa cells already after 24 h incubation. On the other hand, significant toxic effects were detected on MCF7 cells for **3** only after 48 h incubation, with a marked dose-dependent trend observed after 72 h. In contrast, no relevant effect was observed upon treatment with **6** even after 72 h incubation, thus not allowing calculating the IC_50_ value for this compound on these cells. As far as A431 cells are concerned, dose-dependent toxic effects were detected for **3** after 24 and 48 h incubation, whereas lower cytotoxic effects were found after 72 h incubation; in turn, dose-dependent toxic effects were observed for **6** after 48 h incubation. On the basis of the obtained results for both compounds, IC_50_ values were calculated for the data acquired after 48 h incubation (Table 3). IC_50_ values of 25, 31 and 42 μM were obtained for **3** on HeLa, MCF7 and A431 cells, respectively, while IC_50_ values of 32 and 44 μM were obtained for **6** on HeLa and A431 cells, respectively. Overall, with the only exception of **6** on MCF7 cells, both compounds showed a similar antiproliferative activity on the different human cancer cells here tested, with IC_50_ values in the low micromolar range.

## 3. Conclusions

Seven natural compounds, i.e., the dimeric phenanthrenoids **1**, **2** and **3**, the dihydrophenanthrene glucosides **4** and **5**, the tetrahydropyrene glucoside **6** and the 9,10-dihydrophenanthrene **7** were here evaluated as potential binders of G-quadruplex structures of different topologies. The 9,10-dihydrophenanthrene **7** was unable to bind to G-quadruplex structures, in agreement with our previous data on related analogues [14], thus definitively corroborating the finding that the 9,10-dihydrophenanthrenes are not interesting as potential G-quadruplex ligands.

Conversely, dimeric and glucoside derivatives of 9,10-dihydrophenanthrenes proved to bind G-quadruplex structures, particularly the dimeric phenanthrenoid **3** and the tetrahydropyrene glucoside **6**.

Based on the obtained results, by inspection of the chemical structures of the three dimeric phenanthrenoids **1**, **2** and **3**, it can be inferred that the two hydroxyl groups in **3** which are absent in **1** are essential for the binding to both G-quadruplex and duplex structures, while the presence of an additional hydroxyl group (OH-5a) and the lower conformational freedom of **2** compared to **3** seem to be beneficial for the interaction to G-quadruplexes, even though detrimental in terms of G-quadruplex vs. duplex selectivity.

In parallel, by comparison of the chemical structures of the three glucosides **4**, **5** and **6**, it is evident that all the tested 9,10-dihydrophenanthrenes conjugated with a glucose unit are not good G-quadruplex binders. In contrast, if the glucose unit is linked to a pyrene-based scaffold, a higher affinity for the G-quadruplex structures is observed, thus suggesting the major role of the aglycone portion of this class of compounds vs. the sugar in G-quadruplex binding.

Particularly, molecular modelling studies indicated **3** and **6** as binders of the G-quadruplex grooves. This binding mode was indeed consistent with the obtained experimental findings, i.e., the low-to-moderate affinity of the two ligands towards G-quadruplexes, as found in the G4-CPG assay, and the limited effects on the main conformations of the G-quadruplex models produced by the two ligands upon binding, as observed by CD titrations, in agreement with previous reports [11,12,13,14,15,16]. The higher stabilizing effects observed by CD on the parallel oncogenic G-quadruplex model compared to the hybrid telomeric one can be attributed to the binding of **6** in a region of the G-quadruplex groove close to the outer G-quartet, where the ligand can affect the G-quartets stacking. Finally, molecular modelling evidenced a binding preference of both **3** and **6** for the hybrid telomeric G-quadruplex model, which in the case of **3** could be experimentally confirmed by fluorescence titrations.

Remarkably, **3** and **6** showed antiproliferative activity on different human cancer cells with IC_50_ values in the low micromolar range, proving to be promising compounds able to target telomeric and oncogenic G-quadruplex structures and potentially produce anticancer effects.

Taken together, these data showed the two studied natural compounds, **3** and **6**, as valuable leads for the design and synthesis of novel analogues which could result in optimized ligands able to selectively bind a specific G-quadruplex topology and putatively produce strong antitumour effects in the context of targeted anticancer therapies.

## 4. Materials and Methods

### 4.1. G4-CPG Assay

The investigated natural compounds are part of an in-house mini-library of Prof. Marina Della Greca available at the Department of Chemical Sciences of Federico II University of Naples, Italy. The chemical identity of compounds was assessed by re-running NMR experiments and proved to be in agreement with the literature data reported for each compound [17,18,19,20]. The purity of the compounds, checked by reversed-phase High-Performance Liquid Chromatography (HPLC), was always higher than 95%.

Stock solutions (4 mM) were prepared by dissolving a known amount of each compound in pure DMSO. A measured volume was taken from the stock solution to obtain a 60 μM compound solution in 50 mM KCl, 10% DMSO and 10% CH_3_CH_2_OH aq. solution. The detailed general procedure adopted for the G4-CPG assay is here described: weighed amounts of the nude CPG and G-quadruplex-/duplex-functionalized CPG supports (ca. 8 mg) [21] were left in contact with 300 μL of the compound solution in a polypropylene column (4 mL volume, Alltech, Washington, USA) and equipped with a polytetrafluoroethylene frit (10 μm pore size), a stopcock and a cap. After incubation on a vibrating shaker for 4 min, each support was washed with defined volumes of the washing solution (50 mM KCl, 10% DMSO, 10% CH_3_CH_2_OH aq. solution) or the releasing solution (2.5 M CaCl_2_, 15% DMSO aq. solution or pure DMSO) and all the eluted fractions were separately analysed by UV measurements. After treatment with the releasing solution, inducing G-quadruplexes and hairpin duplex denaturation, the supports were suspended in the washing solution and then subjected to annealing, by taking them at 75 °C for 5 min and then slowly cooling to room temperature. The oligonucleotide sequences attached to the supports were: (i) d(TTAGGGTTAGGGTTAGGGTTAGGGTT] (tel_26_), (ii) d(TGGGGAGGGTGGGGAGGGTGGGGAAGGTGGGGA) (c-myc) and (iii) d(CGCGAATTCGCGTTTCGCGAATTCGCG) (ds_27_).

The UV measurements were performed on a JASCO V-550 UV-vis spectrophotometer. A quartz cuvette with a path length of 1 cm was used. The UV quantification of the compounds was determined by measuring the absorbance relative to the λ_max_ characteristic of each compound and referring it to the corresponding calibration curves. The errors associated with the % of bound ligands are within ±2%.

### 4.2. Circular Dichroism

CD spectra were recorded in a quartz cuvette with a path length of 1 cm, on a Jasco J-715 spectropolarimeter equipped with a Peltier-type temperature control system (model PTC-348WI). The spectra were recorded at 20 °C in the range 240–600 nm, with 2 s response, 200 nm/min scanning speed and 2.0 nm bandwidth and were corrected by subtraction of the background scan with buffer. All the spectra were averaged over 3 scans. The oligonucleotides d[(TTAGGG)_4_TT] (tel_26_), d(TGAGGGTGGGTAGGGTGGGTAA) (Pu22) and d(CGCGAATTCGCG) (ds_12_) were purchased from Biomers (Ulm, Germany) as HPLC-purified, desalted compounds with a purity > 99%. The oligonucleotides tel_26_ and ds_12_ were dissolved in a 20 mM KCl, 5 mM potassium phosphate buffer (pH 7), while Pu22 was in a 10 mM Tris-HCl buffer (pH 7), thus obtaining 2 μM solutions, which were then annealed by heating at 95 °C for 5 min, followed by slow cooling to room temperature.

CD titrations were obtained by adding increasing amounts of the ligands (up to 10 molar equivalents, corresponding to a 20 μM solution in ligand) to the oligonucleotide solutions.

For the melting experiments, the CD signal was recorded at 290, 265 and 251 nm for tel_26_, Pu22 and ds_12_, respectively, with a temperature scan rate of 1 °C/min in the range of 10–95 °C.

Particularly, in CD experiments, Pu22 was analyzed in a metal cation-free buffer considering that even in the presence of very low amounts of metal cations the Pu22 G-quadruplex is so stable that its T_m_ value, as well as the related ΔT_m_ values in the presence of each ligand, cannot be accurately determined.

### 4.3. Fluorescence Spectroscopy

Fluorescence spectra were recorded at 20 °C on HORIBA JobinYvon Inc. (Glasgow, UK), FluoroMax^®^-4 spectrofluorometer equipped with F-3004 Sample Heater/Cooler Peltier Thermocouple Drive, by using a quartz cuvette with a 1 cm path length. Excitation wavelengths of 277 and 288 nm were used and the spectra were registered in the range 295–525 and 300–550 nm for **3** and **6**, respectively. The same oligonucleotides described in the subsection of the CD studies were used also for the fluorescence-binding assays.

Titrations were carried out at a fixed concentration (2.0 μM) of **3**. Increasing amounts of tel_26_, Pu22 or ds_12_ (up to 5 μM concentration) were added from 120 μM annealed stock solutions of each DNA sample dissolved in a 20 mM KCl, 5 mM potassium phosphate buffer (pH 7). After each addition, the system was allowed to equilibrate for 10 min before recording the spectra.

The fraction of bound ligand was calculated from the fluorescence intensity at 375 nm and reported in the graph as a function of the DNA concentration. The fraction of the bound ligand was determined using the equation:(1)$$\alpha =\frac{Y-Y_{0}}{Y_{b}-Y_{0}}$$
where *Y*, *Y*_0_ and *Y_b_* are the values of fluorescence emission intensity at the maximum at each titrant concentration, at the initial and final state of the titration, respectively. These points were fitted with an independent and equivalent-sites model using the Origin 8.0 program [32].

The equation of the independent and equivalent-sites model is as follows:(2)$$\alpha =\left(\frac{1}{2{\left[L\right]}_{0}}\right)\left\{\left({\left[L\right]}_{0}+n\left[DNA\right]+\frac{1}{K_{b}}\right)-\sqrt{{\left({\left[L\right]}_{0}+n\left[DNA\right]+\frac{1}{K_{b}}\right)}^{2}-4{\left[L\right]}_{0}n\left[DNA\right]}\right\}$$
where *α* is the mole fraction of the ligand in the bound form, [L]_0_ is the total ligand concentration, [DNA] is the added DNA concentration, n is the number of the equivalent and independent sites on the DNA structure and K_b_ is the binding constant.

### 4.4. Molecular Docking

The NMR deposited structures of the G-quadruplex- and duplex-forming oligonucleotides tel_26_ (PDB 2JPZ), Pu22 (PDB 1XAV) and ds_12_ (PDB 1NAJ) were used. Molecular docking calculations were carried out using AutoDock Vina with the aid of its graphical user interface AutoDockTools [33]. The ligands and DNA targets were prepared by use of AutoDockTools and UCSF Chimera by assigning bond orders, adding hydrogen atoms and generating the appropriate protonation states. The ligands and targets were then converted to proper Autodock PDBQT file formats and the Gaisteiger charges were assigned. The 3D grid box dimensions were defined including the whole DNA macromolecules. The docking area was centred on the DNA centre of mass and grid boxes of 92 Å × 80 Å × 60 Å, 94 Å × 74 Å × 96 Å and 64 Å × 110 Å × 56 Å for tel_26_ G-quadruplex, Pu22 G-quadruplex and ds_12_ duplex, respectively, with a 0.375 Å spacing, were used. 100 docking poses were generated by using a docking parameters seed = random, exhaustiveness = 24 and a number of binding modes = 20 for each of the 5 runs performed for each DNA/ligand system. Docking poses were clustered on the basis of their root-mean square deviation and ranked on the basis of binding energy. Molecular modelling figures were drawn by UCSF Chimera.

### 4.5. Biological Assays

#### 4.5.1. Cell Cultures and Cytotoxicity Assays

Human HeLa CCL-2™ adenocarcinoma cells, MCF7 breast cancer cells and A431 CRL-1555™ epidermoid carcinoma cells were obtained from the American Type Culture Collection (ATCC, Manassas, VA, USA). They were cultured in high-glucose Dulbecco’s modified Eagle’s medium (DMEM) supplemented with 10% fetal bovine serum (FBS), 1% antibiotics (Pen/strep), and 1% L-glutamine at 37 °C in a humidified atmosphere containing 5% CO_2_. To perform cytotoxicity assays, cells were seeded into 96-well plates at a density of 3 × 10^3^ cells/well. After 24 h, the cell supernatant was replaced with a fresh medium containing increasing concentrations of compounds under test. Following 48 h incubation, MTT assays were performed as previously described [34]. Briefly, cell culture supernatants were replaced with 0.5 mg/mL MTT reagent dissolved in DMEM medium without red phenol (100 μL/well). After 4 h of incubation at 37 °C, the resulting insoluble formazan salts were solubilized in 0.01 M HCl in anhydrous isopropanol and quantified by measuring the absorbance at 570 nm by using an automatic plate reader spectrophotometer (Synergy H4 Hybrid Microplate Reader, BioTek Instruments, Inc., Winooski, VT, USA). Cell viability was expressed as means of the percentage values obtained by comparison with control untreated cells.

#### 4.5.2. Statistical Analyses

Statistical analyses were performed by using a Student’s *t*-Test. Significant differences were indicated as * *p* < 0.05.

## Figures and Tables

**Figure 1 ijms-24-07765-f001:**
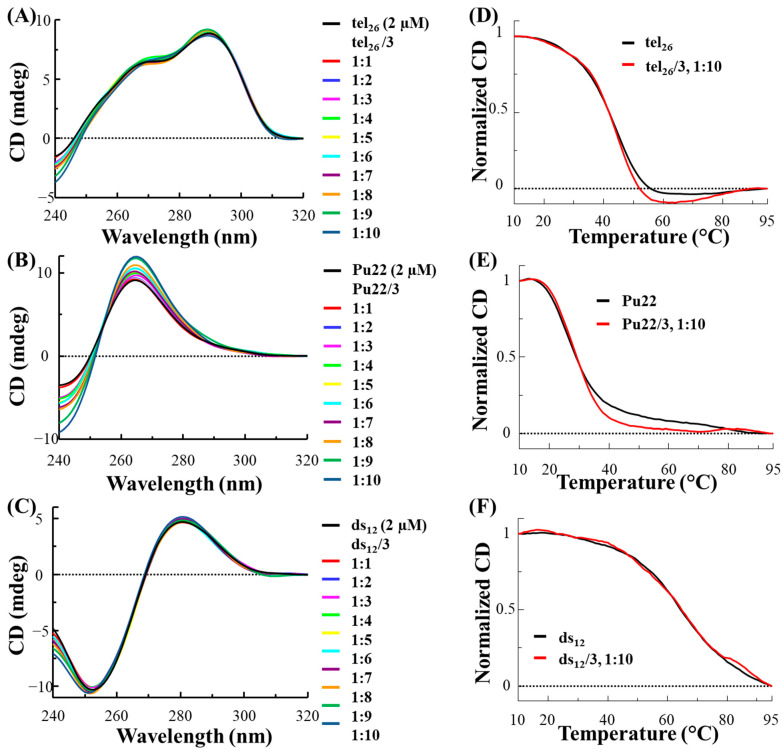
CD spectra of 2 μM solutions of tel_26_ G-quadruplex (**A**), Pu22 G-quadruplex (**B**) and ds_12_ duplex (**C**), in 20 mM KCl, 5 mM potassium phosphate buffer (pH 7) for tel_26_ and ds_12_ or in 10 mM Tris-HCl buffer (pH 7) for Pu22, in the presence of increasing amounts (up to 10 equivalents) of **3**. CD melting curves for tel_26_ G-quadruplex (**D**), Pu22 G-quadruplex (**E**) and ds_12_ duplex (**F**) in the absence and presence of 10 molar equivalents of **3**, recorded at 290, 265 and 251 nm for tel_26_, Pu22 and ds_12_, respectively.

**Figure 2 ijms-24-07765-f002:**
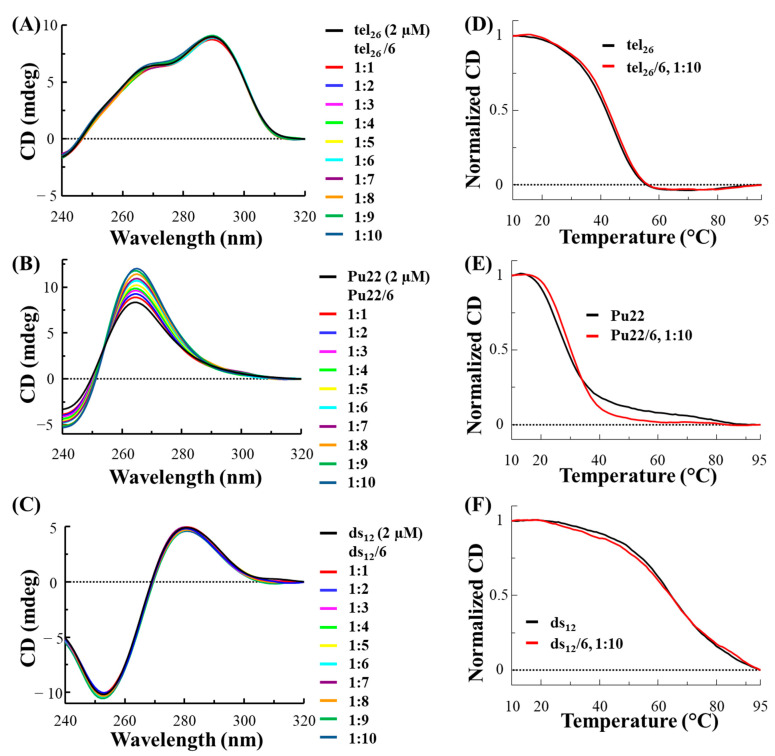
CD spectra of 2 μM solutions of tel_26_ G-quadruplex (**A**), Pu22 G-quadruplex (**B**) and ds_12_ duplex (**C**), in 20 mM KCl, 5 mM potassium phosphate buffer (pH 7) for tel_26_ and ds_12_ or in 10 mM Tris-HCl buffer (pH 7) for Pu22, in the presence of increasing amounts (up to 10 equivalents) of **6**. CD melting curves for tel_26_ G-quadruplex (**D**), Pu22 G-quadruplex (**E**) and ds_12_ duplex (**F**) in the absence and presence of 10 molar equivalents of **6**, recorded at 290, 265 and 251 nm for tel_26_, Pu22 and ds_12_, respectively.

**Figure 3 ijms-24-07765-f003:**
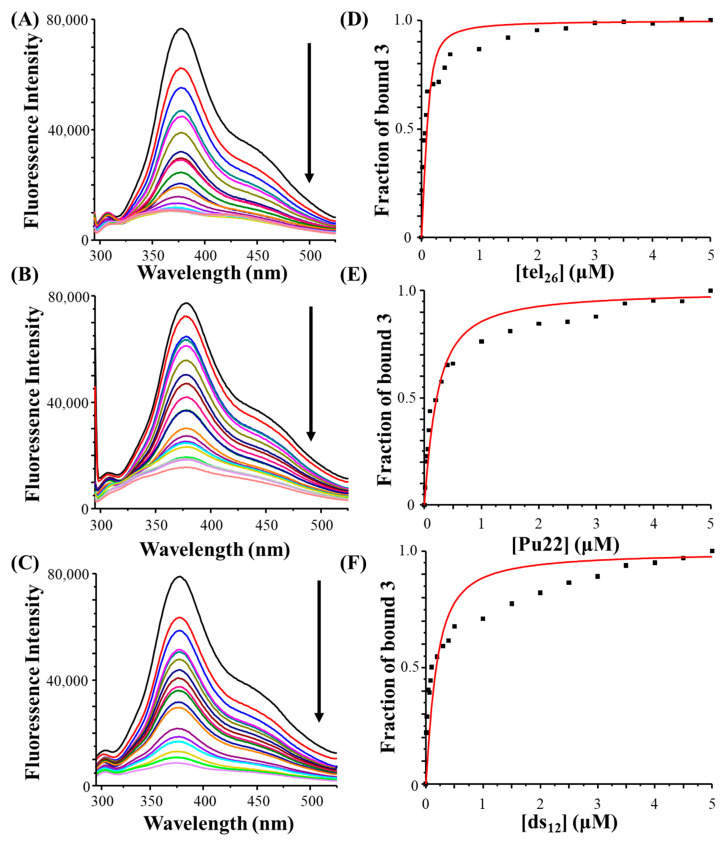
Left panels: Fluorescence emission spectra obtained by adding increasing amounts of tel_26_ G-quadruplex (**A**), Pu22 G-quadruplex (**B**) and ds_12_ duplex (**C**) to 2 μM solutions of **3** (black lines). Arrows indicate the variation of the fluorescence intensity on increasing DNA concentration. Right panels: Representative binding curves obtained by plotting the fraction of bound **3** to tel_26_ G-quadruplex (**D**), Pu22 G-quadruplex (**E**) and ds_12_ duplex (**F**) as a function of the DNA concentration. The black squares represent the experimental data; the red line represents the best fit obtained using an independent and equivalent-sites model.

**Figure 4 ijms-24-07765-f004:**
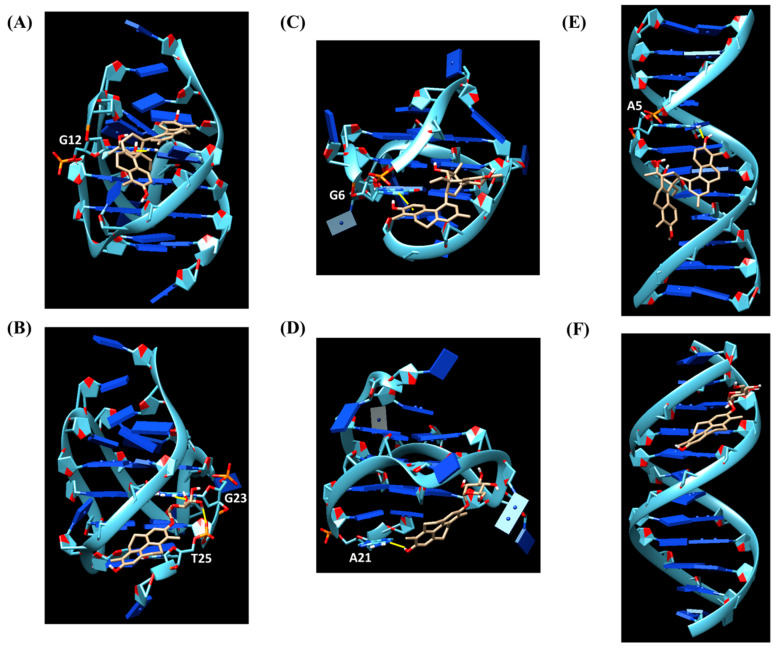
Binding modes of **3** and **6** when docked to tel_26_ G-quadruplex ((**A**) and (**B**), respectively), Pu22 G-quadruplex ((**C**) and (**D**), respectively) and ds_12_ duplex ((**E**) and (**F**), respectively). Ligands are represented as beige sticks. Hydrogen bonds are shown as yellow bold lines, while π-π interactions as yellow dashed lines.

**Figure 5 ijms-24-07765-f005:**
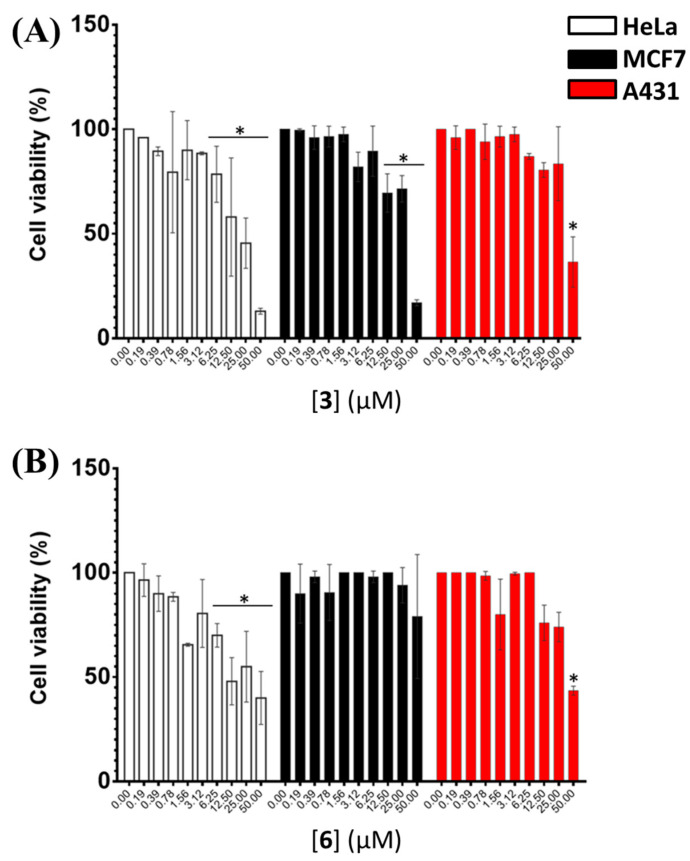
Effects of increasing concentrations of (**A**) **3** and (**B**) **6** (0–50 μM) on the viability of HeLa, MCF7 and A431 human cancer cells after 48 h of incubation. Cell viability values are expressed as the percentage of viable cells for treated vs. control cells grown in the absence of the tested compounds. Three independent experiments were performed with triplicated determinations. * *p* < 0.05 were obtained for treated vs. control samples.

**Table 1 ijms-24-07765-t001:** Chemical structures of the investigated natural compounds. Atoms and ring numbering are reported only for the atoms cited in the text.

Family	Compound	Chemical Structure
Dimeric dihydrophenanthrenoids	**1**	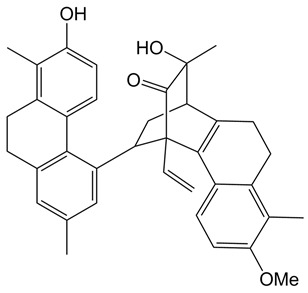
	**2**	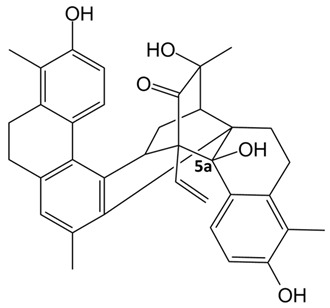
	**3**	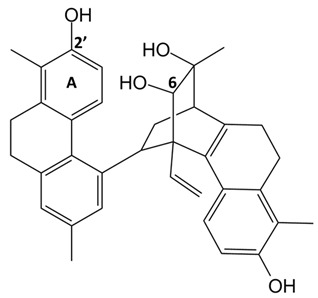
Dihydrophenanthrene glucosides	**4**	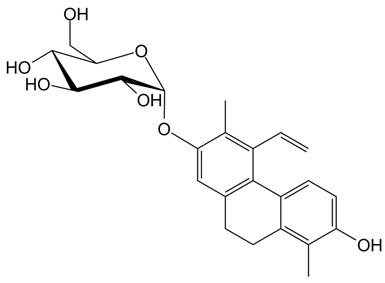
	**5**	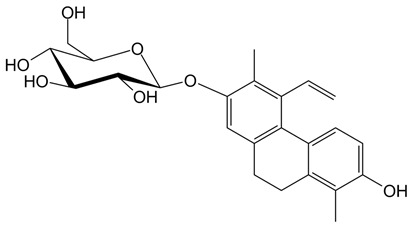
Tetrahydropyrene glucosides	**6**	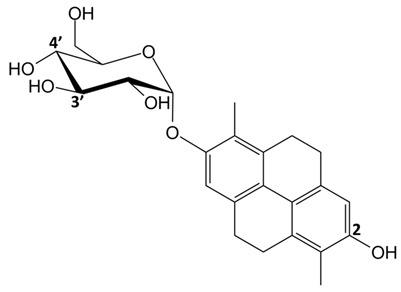
Dihydrophenanthrenes	**7**	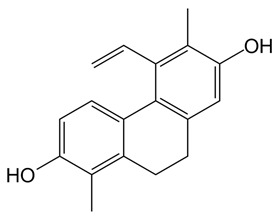

**Table 2 ijms-24-07765-t002:** Summary of the binding assay data obtained for the natural compounds by the G4-CPG assay. The amounts of bound ligand are calculated as a difference between the initially loaded amount of ligand and the unbound ligand, recovered by washings with the solution 50 mM KCl, 10% DMSO, 10% CH_3_CH_2_OH, and are expressed as percentage of the quantity initially loaded on each support. The errors associated with the reported percentages are typically within ±2%.

Compound	Bound Ligand (%)		
	CPG-tel_26_	CPG-c-myc	CPG-ds_27_
**1**	0	0	0
**2**	48	56	38
**3**	19	18	0
**4**	0	0	0
**5**	0	3	0
**6**	9	8	2
**7**	2	4	0

**Table 3 ijms-24-07765-t003:** IC_50_ values were determined by testing increasing concentrations of each compound for 48 h on HeLa, MCF7 and A431 human cancer cells by MTT assays. N/A: not applicable.

	IC_50_ (μM)		
Compound	HeLa	MCF7	A431
**3**	**25**	**31**	**42**
**6**	**32**	**N/A**	**44**

## Data Availability

Not applicable.

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
