# Peer review of "Exploring the Binding of Natural Compounds to Cancer-Related G-Quadruplex Structures: From 9,10-Dihydrophenanthrenes to Their Dimeric and Glucoside Derivatives"

_ijms, 2023, doi:10.3390/ijms24097765_

Round 1

Reviewer 1 Report

This study is typical of its kind. The experiments seem to be performed carefully, and conclusions are clear, although are not spectacular. 

Reviewer 2 Report

In present time search for anti-cancer natural compound is highly emerging. Authors identified G quadruplex binders which will be an interesting topic for researchers.

The article can be accepted after some minor corrections

1. Lattin word shuld be in italic form.

2. Authors should check the structures given in table 1. 

3. Authors should explain why different buffer is used for pu22.

4. Authors should study the binding phenomenon UV vis spectroscopy also. 

Reviewer 3 Report

With the final goal of discovering anti-cancer compounds from natural compounds, the authors searched for seven phenanthrenoids with affinity for G-quadleplexes at first. Two of which showed their affinities for immobilized G-quadleplexes and the affinity of a phenanthrenoid was determined by fluorescent titration. Theoretical calculations of molecular docking were also performed, and growth inhibition was observed in three cancerous cell types in culture.

This research is at the stage of observing physical properties and the killing effect on cultured cells, but does not include medical examinations. Therefore, as a work stimulating the further research, it is necessary to conduct as much analysis as possible in physical properties.

1.    A weakness of this paper in the analysis of the physical properties is that the binding constant of compound 6 was not determined in the fluorescence measurements simply because of the reason that compound 6 does not fluoresce. In principle, it may be possible to determine the binding constant of analogous compound 3 by fluorescence measurement because it may compete with analogous compound 3, which emits fluorescence. If such a method is not possible, at a minimum, the reason should be written. 

2.    Although fluorescent titration is quantitatively quite precise, a systematic, one-to-one deviation from binding is observed in the titration curve of compound 3 in Figure 3. Possible causes for this should be discussed. In particular, the formation of the compound's own aggregates and multiple binding may be pharmacologically important, such as the need for a carrier. However, this is not discussed at all, but only dogmatically and ambiguously stated "All compounds proved to be fully soluble and stable in the assay solutions without showing any evidence. 

3.    In the measurement of cytotoxicity, the interpretation is not objective enough. Even under a single condition, the IC50 concentrations for Hela and A431 are comparable, contradicting the authors' conclusion that Hela is preferentially inhibited in growth. The results are insufficient at the cell biological level unless experiments are conducted under more various conditions, preferably including a control of non-cancer-derived cells.

As a basic study that should give hints for further development, the manuscript lacks convincing experiments and more detailed descriptions of physical properties and cytotoxicity.

This manuscript contains several weak points to be revised before publication.

Round 2

Reviewer 3 Report

With the final goal of discovering anti-cancer compounds from natural compounds, the authors searched for seven phenanthrenoids with affinity for G-quadleplexes at first. Two of which showed their affinities for immobilized G-quadleplexes and the affinity of a phenanthrenoid was determined by fluorescent titration. Theoretical calculations of molecular docking were also performed, and growth inhibition was observed in three cancerous cell types in culture.

This research is at the stage of observing physical properties and the killing effect on cultured cells, but does not include medical examinations. Therefore, as a work stimulating the further research, it is necessary to conduct as much analysis as possible in physical properties.

1.    A weakness of this paper in the analysis of the physical properties is that the binding constant of compound 6 was not determined in the fluorescence measurements simply because of the reason that compound 6 does not fluoresce. In principle, it may be possible to determine the binding constant of analogous compound 3 by fluorescence measurement because it may compete with analogous compound 3, which emits fluorescence. If such a method is not possible, at a minimum, the reason should be written. 

2.    Although fluorescent titration is quantitatively quite precise, a systematic, one-to-one deviation from binding is observed in the titration curve of compound 3 in Figure 3. Possible causes for this should be discussed. In particular, the formation of the compound's own aggregates and multiple binding may be pharmacologically important, such as the need for a carrier. However, this is not discussed at all, but only dogmatically and ambiguously stated "All compounds proved to be fully soluble and stable in the assay solutions without showing any evidence. 

3.    In the measurement of cytotoxicity, the interpretation is not objective enough. Even under a single condition, the IC50 concentrations for Hela and A431 are comparable, contradicting the authors' conclusion that Hela is preferentially inhibited in growth. The results are insufficient at the cell biological level unless experiments are conducted under more various conditions, preferably including a control of non-cancer-derived cells.

As a basic study that should give hints for further development, the manuscript lacks convincing experiments and more detailed descriptions of physical properties and cytotoxicity.

Author Response

See attached review
